# Precision and Accuracy of Field Versus Laboratory Bioassay Insecticide Efficacy for the Control of Immature *Bemisia tabaci*

**DOI:** 10.3390/insects14070645

**Published:** 2023-07-18

**Authors:** Paulo S. G. Cremonez, Jermaine D. Perier, Mirela M. Nagaoka, Alvin M. Simmons, David G. Riley

**Affiliations:** 1Department of Entomology, University of Georgia Tifton Campus, Tifton, GA 31793, USA; jermaine.perier@uga.edu (J.D.P.); mirela.nagaoka@uga.edu (M.M.N.); dgr@uga.edu (D.G.R.); 2U.S. Vegetable Laboratory, Agricultural Research Service, USDA, 2700 Savannah Highway, Charleston, SC 29414, USA; alvin.simmons@usda.gov

**Keywords:** insecticide resistance management (IRM), maximum-dose bioassay, sweetpotato whitefly, *Bemisia tabaci*, insecticide response

## Abstract

**Simple Summary:**

Measurements of experimental errors between controlled laboratory bioassays and field response of insecticide on immature stages of the sweetpotato whitefly were investigated using parallel sample populations. The bioassays were more precise in measuring insecticide efficacy compared to homologous field trials. Seasonal variations in precision and accuracy indicate the importance of identifying and considering external confounding factors when estimating insecticide efficacy or predicting field response through bioassays. These results highlight the value and limitations of using quick, viable, and easy-to-set-up bioassays to quantify the efficacy of commonly used products on target pest species, particularly those commonly associated with insecticide resistance, using the sweetpotato whitefly as a model.

**Abstract:**

Ecotoxicological studies often result in reports on the limitation and sometime failures of biological assay data to predict field response to similar treatments. Nevertheless, it is widely accepted that controlled bioassays can better quantify the specific mortality response of a target pest species to a specific toxin. To quantify the relationship between whitefly bioassay and field response data, we evaluated a controlled laboratory bioassay and a concurrent cucurbit field trial method to assess insecticide efficacy for controlling the sweetpotato whitefly, *Bemisia tabaci* (Gennadius) (Hemiptera: Aleyrodidae). This was based on oviposition and nymphal development. We specifically tested the assumptions that a maximum dose bioassay would more precisely measure insecticide efficacy as compared with a comparable field spray test evaluation, and the response would be equal between the bioassay and the field as a measure of control accuracy for both adult oviposition and development of nymphal stages. To make a direct comparison, we tested the same whitefly population subsamples from 352 plots in eight cucurbit field experiments in Georgia, USA, in 2021 and 2022. The bioassays provide significantly precision for estimating proportional whitefly response. As expected, treatment-specific nonequivalence in immature whitefly counts between the bioassay and field, i.e., a lack of accuracy, only occurred with insecticides that were not highly toxic to all growth stages of whiteflies.

## 1. Introduction

The foundation of toxicological research with insecticides relies on controlled laboratory bioassays that typically result in lethal dose (LD) or lethal concentration (LC) estimates for specific toxins used to control specific pest insect species [1]. The standard for evaluating insecticide efficacy in the field is replicated spray tests, often using the highest labeled rate to assess the maximum control achievable for a specific toxin. A maximum dose laboratory bioassay was recently proposed for whiteflies as a less costly, more efficient alternative to assessing insecticide efficacy in field populations than multiple spray tests [2]. Ecotoxicological studies have often indicated that predicting field response based on a laboratory bioassay can have many problems [3,4]. Therefore, we investigated whether a laboratory insecticide bioassay for whitefly control precisely and accurately estimates field whitefly response to the insecticide.

The sweetpotato whitefly, *Bemisia tabaci* (Gennadius) (Hemiptera: Aleyrodidae), is an important, polyphagous agricultural pest that causes significant economic losses worldwide in many crops [5]. This pest feeds on multiple vegetables, agronomic crops, fruits, and ornamental plants, often leading to significant yield losses and decreased crop quality [6]. Insecticide application is a common practice for its control; however, developing insecticide resistance is becoming an increasing concern for growers [7]. Therefore, it is crucial to evaluate the efficacy of insecticides against *B. tabaci* to ensure effective pest and insecticide resistance management (IRM) strategies.

Evaluating insecticide efficacy can be challenging due to the potential statistical errors associated with sampling and other sources of data variability in a farmscape compared with the relatively homogeneous conditions in a laboratory bioassay. To obtain a truly representative efficacy estimate, variables contributing to confounding factors must be eliminated or minimized, including gross, systematic, and random errors. Gross errors (blunders) and systematic errors (biases), although unpredictable, can be removed or minimized in practice, while random errors arise from intrinsic factors and arbitrary observational errors [8,9]. By identifying and minimizing blunders and biases (interfering factors), one can approach an optimal estimate of efficacy by accounting for random errors [10]. Evaluating these factors is critical to ensure optimal insecticide efficacy estimates and predictability in field response.

The efficacy of several insecticides has already been determined for adult whiteflies from populations collected in South Georgia, USA [11]. In that study, a 24-h laboratory test was efficiently correlated with the field numbers resulting from a concurrent application throughout eight cucurbit field trials during the summers of 2021 and 2022. This rapid, viable, and easy-to-setup bioassay proved to be a potential tool for determining insecticide efficacy fluctuations in a given whitefly population. However, some commonly used insecticides with different modes of action may need to be evaluated differently due to their mode of action, such as the insect growth regulator pyriproxyfen [12,13,14]. Moreover, the target effects of many insecticides affect not only the adult populations and, therefore, oviposition, but also the establishment of subsequent immature stages. Determining insecticide efficacy on whitefly immatures takes more response time, which in the farmscape can lead to increasing levels of confounding factors that can obscure the measure of chemical efficacy. We can use precision in both field and laboratory tests to provide a direct assessment of experimental error and evaluate the accuracy of the laboratory results relative to field results to begin a discussion on what each measure of efficacy is actually being reported.

The objectives of this study were (1) to evaluate the precision of field and laboratory measurements of the efficacy of selected insecticides, (2) to determine the accuracy of the results based on eggs and nymphs of *B. tabaci* assessed in laboratory conditions as an estimator of efficacy in the field using a standard spray methodology, and (3) to identify the potentially crucial sources of errors involved in field evaluations. By addressing these objectives, we hope to provide valuable insights into the value of widely used standard efficacy evaluations used by the pesticide industry for decades, especially for whitefly control. Our working hypothesis is that a whitefly bioassay provides comparable quality data to a field evaluation but at a fraction of the cost. The bioassay tools used here could lead to more reliable and effective whitefly IRM programs than annual field efficacy studies.

## 2. Materials and Methods

### 2.1. Field Trials

Field trials were conducted as described in our previous study on adult efficacy determination [11]: four field trials were conducted at the University of Georgia Coastal Plain Experiment Station in Tifton, GA (31°30′53″ N, 83°32′51″ W farm site), during the summers of 2021 and 2022, to assess the efficacy of commonly used insecticides from different IRAC classes as described in Table 1. The trials were conducted on two crops, squash (*Cucurbita pepo* L.) var. ‘Yellow Crookneck’, and cucumber (*Cucumis sativus* L.) var. ‘Straight 8’, with two trials per crop per year. The test plants were directly seeded between late June and early August per year, coinciding with the historically annual increase in whitefly populations in Georgia [15]. For these trials, we aimed to evaluate the performance of different insecticides against whiteflies in the field and provide valuable information for growers to make informed decisions regarding pest management. We used a randomized complete block design with four replicates for each crop, and each treatment plot consisted of two rows measuring 18.3 m long × 1.8 m wide. The field treatments were applied using a tractor-mounted pressurized air sprayer at 413.7 kPa (60 psi), with three TX-18 hollow cone tips per row (one overhead and two to the sides) and a total spray volume of 496 L ha^−1^. This application setup used on seedlings (3–5 leaf stage) resulted in heavy soaking of the foliage. The insecticides used in the field and bioassay treatments are summarized in Table 1.

### 2.2. Laboratory Bioassays

Concurrent with the field trials, laboratory bioassays were performed to evaluate the response of whiteflies to the same insecticides used in the field. Bioassays were carried out on whitefly populations (*Bemisia tabaci*, Middle East Asia Minor 1 cryptic species) collected from the same field test plots using insect-free cotton seedlings as the host plant test media. The cotton seedling-tube system method was used as previously described [16]. Cotton seedlings with at least one mature true leaf (4–5 cm width) were carefully removed from the soil medium, with only one true leaf left (tested substrate) attached to each plant. The leaves were dipped in the insecticide mixture and allowed to air dry at room temperature (25 ± 5 °C) to simulate the field application. The roots of each plant were enclosed in a scintillation vial (containing 20 mL of tap water) containing cotton wool, and the vial was sealed with Parafilm^®^ to keep the plant alive during the evaluation period. For the bioassays, proportional treatment concentrations of the per-hectare rate (as listed in Table 1) were used in the equivalent of 935 L ha^−1^ (equiv. 100 gal. acre^−1^) water spray volume as the maximum labeled dose.

A random sample of approximately 50 unsexed adult *B. tabaci* of mixed ages was collected in ClearTec^®^ tubes (V = 130 mL, ClearTec Packaging, Park Hills, MO, USA), being considered an experimental unit. These tubes, each serving as an individual experimental unit, were modified with side perforations, which were sealed with nylon chiffon fabric to provide ventilation. The collected samples were immediately transported back to the laboratory prior to the field application. Tubes containing field-collected whitefly adults were affixed to previously treated cotton seedling tube systems with a tube sleeve, allowing free access to the abaxial leaf surface. The tubes were maintained under controlled conditions (25 ± 2 °C, 60 ± 5% RH, 24 L:0 D h photoperiod). After 48 h, the remaining adult whiteflies were carefully extracted from the tubes and leaves, and the plants were maintained under the same controlled conditions for an additional seven days.

### 2.3. Data Collection and Analysis

A field insecticide application was carried out simultaneously, after 24 h of the laboratory bioassay, and five random leaf samples (from the third leaf node) per plot were collected to count eggs and nymphs using a stereoscopic microscope and guided with a six-punch hole card (total area = 12.25 cm^2^) covering the leaf. Total *B. tabaci* egg and nymph counts were obtained from the leaves collected on day seven. The bioassay results were compared with standard field scout counts by treatment plots to determine the precision and accuracy of immature (egg and nymph) whitefly numbers in both situations. Because the bioassay and field efficacy data were estimates of the insecticide response for effectively the same whitefly population samples, correlations between field and bioassay responses were based on the estimated efficacy coefficient (EC), whose product is a proportional value between 0 and 1 as described by the formula:$${EC}_{i}=\frac{x_{i}}{x_{max}}$$
where $x_{i}$ is the observed value, and $x_{max}$ is the highest value observed in the trial/bioassay. The precision of the laboratory measurements was evaluated using GLM analysis followed by Tukey‘s ad hoc test (*p* < 0.05) to compare the means of each treatment.

Precision variables were calculated as coefficient of variation (CV) to provide information on the variability and consistency of the data from both field and bioassay estimates. To evaluate the accuracy of the estimates, the bioassay ECs were considered as predicted values, as all surrounding factors other than the tested treatments interfered with the predictive nature of the insecticide efficacy response. Consequently, the field ECs were considered observed values, representing on-farm efficacy, with all uncontrollable factors, such as climate and ecosystem. With this definition, two parameters are commonly used to evaluate the accuracy of continuous predictions, such as the number of surviving insects: the mean absolute error (MAE), defined as the average of the absolute difference between the predicted (laboratory bioassay) and observed values (field trials) for each treatment, and the root mean square error (RMSE), measured as the average distance between the observed and predicted values [17]. Lower values of both MAE and RMSE are indicative of higher accuracy in the laboratory bioassay’s predictions of its corresponding field responses. All trials and bioassay data were analyzed using the SAS^®^ Enterprise Guide v. 8.3 (SAS Institute Inc., Cary, NC, USA) using the PROC GLM procedure to analyze variance.

## 3. Results

Insecticide efficacy response was very similar based on laboratory and field estimates for egg (Figure 1) and nymph count estimates (Figure 2). Consistently, the insecticide treatments presented lower numbers of B. tabaci eggs and nymphs in squash and cucumber from 2021 and 2022 summer field trials than their associated laboratory bioassays. Neurotoxic-based insecticides, such as dinotefuran (4A), flupyradifurone (4D), acetamiprid (4A), and imidacloprid (4A) resulted in higher efficacy on immature control compared to other modes of action. Cyantraniliprole (28), a muscular Ca^2+^ pump inhibitor, was also similarly effective. Sulfoxaflor (4C) resulted in inconsistent efficacy compared to the other neurotoxic compounds. However, it always showed significant egg and nymph reduction compared to untreated plots. Clothianidin (4A), flonicamid (9C), and spiromesifen (23) revealed low efficacy on immature whiteflies. Pyriproxyfen (7C) also revealed low efficacy for egg numbers. However, a higher reduction in nymphal counts was observed for this growth regulator (Appendix A).

Regarding precision analysis, CV values for egg and nymph counts were consistently higher in the field values compared to the bioassays, indicating a more precise characterization of the laboratory estimates (Appendix A). Throughout the study, from 2021 to 2022, field and laboratory precision increased (Figure 3a,b); 2021 field experiments egg counts presented an average variation on CV from 51.5% (squash #2) to 123.6% (squash #1), and nymphs from 78.5% (squash #2) to 129.4% (squash #1). The bioassay egg counts CV varied from 28.0% (cucumber #2) to 66.5% (cucumber #1), and nymphs from 33.5% (cucumber #2) to 86.7% (cucumber #1). The 2022 CV values for egg counts in the field varied from 34.3% (squash #3 and cucumber #4) to 39.7% (squash #4), and nymphs from 20.0% (cucumber #3) to 89.7% (squash #4). The laboratory egg estimates CV varied from 20.8 (squash #3 and cucumber #4) to 38.8% (cucumber #3).

Accuracy analysis presented an inverse trend from the precision results, decreasing throughout the experimental time analyzed, i.e., RMSE values were lower in 2021 than in 2022 for both egg (Figure 3a) and nymph count estimates (Figure 3b). 2021 eggs average accuracy measured through RMSE varied from 0.150 (squash #1) to 0.229 (cucumber #2), and nymphs RMSE values varied from 0.145 (squash #1) to 0.312 (cucumber #1). In 2022, the measured average accuracy in terms of RMSE for egg survival estimate varied from 0.237 (cucumber #4) to 0.357 (squash #3), and nymph RMSE values varied from 0.238 (cucumber #4) to 0.518 (squash #3) (Appendix A).

## 4. Discussion

### 4.1. Insecticide Efficacy and Immature Control

The variation in *B. tabaci* egg numbers after insecticide treatment is a direct effect of the treatment on the adult population. Moreover, changes in the counted nymph population are a direct effect of the insecticide’s activity on egg and early nymphal stages. Studies have been carried out to develop bioassays to define insecticide efficacy on whitefly field populations [2,11]. These studies have only considered survival by direct adult count, but indication of the insecticide effects over immatures were not reported. *B. tabaci* egg and nymph counts in our study correspond to insecticide treatment effect over adult survival and residual effect over egg viability. Additionally, the efficacy of the same group of insecticides over adult whitefly survival was also examined, indicating significant positive correlations between the two environments and suggesting the potential for rapid and easy-to-set-up bioassays to accurately reflect real insecticide efficacy [11]. Studies on adult survival and immature development are critical for understanding population dynamics, especially in the context of how insecticides affect pest establishment and dispersal, and are, therefore, vital for establishing effective integrated pest management (IPM) strategies [18]. In the case of *B. tabaci*, developing control tactics with different classes of insecticides is particularly crucial during initial crop installation, as many whiteflies are migrating from adjacent crops and non-cultivated hosts [19]. Because many adult *B. tabaci* start their establishment on the crop at this time, highly active immature development quickly leads to population build-up, causing significant economic loss. Optimizing insecticide efficacy by targeting early-stage development is essential to mitigate population growth.

### 4.2. Precision and Accuracy

The accuracy analysis of the conducted experiments presented an inverse trend from the precision results. Notably, the RMSE values were lower in 2021 than in 2022 for *B. tabaci* immatures, indicating that the 2021 season presented higher accuracy for estimating field egg (Figure 3a) and nymph (Figure 3b) insecticide response. (Figure 3c). Overall, the insecticide treatments exhibited substantial variation in the CV values for laboratory estimates, indicating significant fluctuations in precision. However, the untreated control consistently displayed a lower CV, signifying greater precision. Meanwhile, the RMSE (and MAE, Appendix A) values remained conserved and generally below 0.5, with only a few exceptions, signifying that the bioassays largely provided accurate predictions of field performance.

To measure the precision and accuracy of an experiment it is necessary to quantify the random errors (variation between repetitions of the same treatment) and uncertainty (distance between observed and expected estimates), respectively. Experiments are generally subjected to gross, systematic, and random errors. The first two are controllable and, although unpredictable, can be removed or minimized in practice: blunders or gross errors are mainly unpredictable and undesired interference caused by blunders, abiotic factors, and human error; systematic errors (or biases) are, by definition, the components that differ the independent (observed) and dependent (expected) variables constantly and proportionally [9]. In contrast, random errors are part of uncontrollable fluctuations between these variables and arise from intrinsic factors of the object of study (e.g., organisms being evaluated) and arbitrary observational errors [8]. Therefore, by identifying and eliminating or minimizing the blunders and systematic errors (interfering factors), one can approach an optimal estimate of efficacy by measuring random errors (precision) and the difference between the expected (estimated) and observed (true) measurements from a test [10].

Precise and accurately measuring insect populations is critical for effective pest management. In this study, we evaluated the precision and accuracy of laboratory bioassays for determining the numbers of treated *B. tabaci* eggs and nymphs compared to field data. Theoretically, the factors interfering in the field are reduced or minimized by importing only the whitefly population (experimental object) and the insecticide treatments (independent variables) to the laboratory bioassay. Thus, assuming the predictability of the bioassay, measuring accuracy is possible by assuming the efficacy estimate from the laboratory as the estimated variable (ŷ) and the field estimate as the observed variable (y). In this study, it was possible to observe a relationship between field and bioassay precision estimates, and its resulting accuracy measure, as an interdependent response (Figure 3).

### 4.3. Predictability Power

Precision and accuracy are fundamental forecasting parameters measuring inter-treatment (covariance) random errors and differential responses from estimated and observed results. How good the estimated (laboratory) response will be is based on how fair and clean the true (field) estimate was taken in the first place. Other factors that might interfere with the increase and decrease of precision and accuracy might include inter-relationships of field factors and blunders with the experimental variable of interest (e.g., whitefly population). Additionally, the parental populational size and reintroduction factors are intrinsically on the variable of interest, as they might pose a differential response over time (Figure 1). These factors are variable and dynamic and continue to occur in the field, interfering with the data over time. Thus, the quick-response analysis (24 h) was crucial to approach the predictability of the bioassay component.

In addition to serving as the fundamental basis for gauging model errors, precision and accuracy are extensively used in numerous forecasting and model performance validation works [17,20,21]. They are basic statistical metrics to measure model errors, with lower Coefficient of Variation (CV) and Root Mean Square Error (RMSE) values indicating higher precision and accuracy estimates, respectively, and a better fit of the model under evaluation. The straightforward application of these parameters is adequate for validating the model used and provides important insight whether a strong correlation is concise with the representability of the analyzed parameters. These predictive measurements hold considerable potential for aiding growers in decision-making processes regarding whitefly infestation levels, enabling proactive action before reaching economic loss thresholds.

### 4.4. Viability of Pre-Treatment Bioassay Sampling

Laboratory bioassays have become the standard method for evaluating the efficacy of insecticides and monitoring the development of resistance in pest populations. The pre-treatment bioassays used in this study provided precise and accurate estimates of insecticide efficacy against *B. tabaci*, focusing on key developmental stages such as the egg and nymph. Furthermore, these bioassays deliver quick results, making them a valuable tool for crop pest specialists and growers in whitefly control. Although laboratory bioassays are more precise due to their controlled environment, it is essential to consider the viability of using field trial sampling concurrently with laboratory trials to account for the complexity of active factors in the real-world setting. Despite these limitations and assuming a presumable weak forecasting power, the advantages of using bioassays as a pre-treatment predictive-approach tool are based on their relatively low economic cost, easy-to-setup functionality, and quick results delivery. This raises important questions about the scalability of bioassays, as well as the potential for confounding factors and variations in field conditions that could affect their effectiveness.

To determine the validity of using a laboratory bioassay as a predicting-approach tool for insecticide efficacy of *B. tabaci* field populations, we considered the premises of statistical models for experimentation, with the dependent variable being insecticide efficacy estimated by egg and nymph relative count (0.00–1.00), i.e., the higher the estimate, the lower the treatment efficacy. We assumed that the field agroecosystem complex represents the real world, and the trials resulted in the actual (observed) data. Furthermore, the controlled environment-based laboratory bioassay represents the hypothetical condition, and the predictive (expected) data were measured. These two different environments share two common key factors: the object of study (whitefly population) and the independent variable (insecticide treatments). By importing the key factors from the field to the bioassay, we can create a model for comparison and analyze precision and accuracy with a valid model. However, it is important to determine the factors pertinent to the field environment that significantly impact the differences between the observed and expected estimates. Understanding these factors is critical to ensuring that the results obtained from the bioassay are reliable and can be used for decision-making in the field. A schematic idea of the interfering factors based on this study is presented in Figure 4.

In general, laboratory bioassays are of different types [16,22,23]. As reported by other studies, they may serve as an important quick test to determine insecticide susceptibility loss and aid decision-making in field situations [2,24,25,26,27]. In a previous study [11], we observed a high correlation between laboratory bioassays and the field efficacy of several insecticides over adult *B. tabaci*. The results obtained in our current work build upon this previous finding, demonstrating that precise and accurate bioassays can vary in field conditions permeated with interfering factors, such as whitefly population reintroduction. Our findings highlight the potential of laboratory bioassays to quickly predict an insecticide resistance surge in whitefly populations.

## 5. Conclusions

Our results demonstrated that field trials and laboratory bioassays help determine insecticide efficacy based on *B. tabaci* immature counts. Bioassays were more precise than the field methods. However, they are a conceptualized idea of the real world as interfering factors are removed. These findings provide valuable information for developing effective whitefly management strategies and highlight the importance of considering multiple factors when evaluating the efficacy of insecticides in the field based on laboratory bioassay results.

## Figures and Tables

**Figure 1 insects-14-00645-f001:**
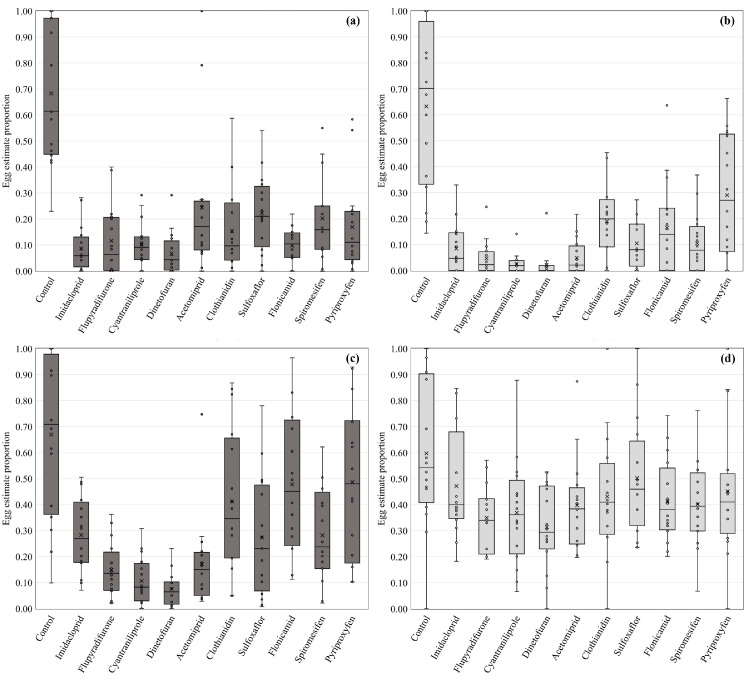
Proportional estimate of *Bemisia tabaci* egg numbers from field and laboratory insecticide efficacy trials. (**a**) field estimate from the 2021 season; (**b**) laboratory bioassay estimate from the 2021 season; (**c**) field estimate from the 2022 season; (**d**) laboratory bioassay estimate from the 2022 season. The box plot includes the interquartile range (IQR, the box), the median (line in the box), data variability (whiskers), overall mean (X), and trial/bioassay means (dots).

**Figure 2 insects-14-00645-f002:**
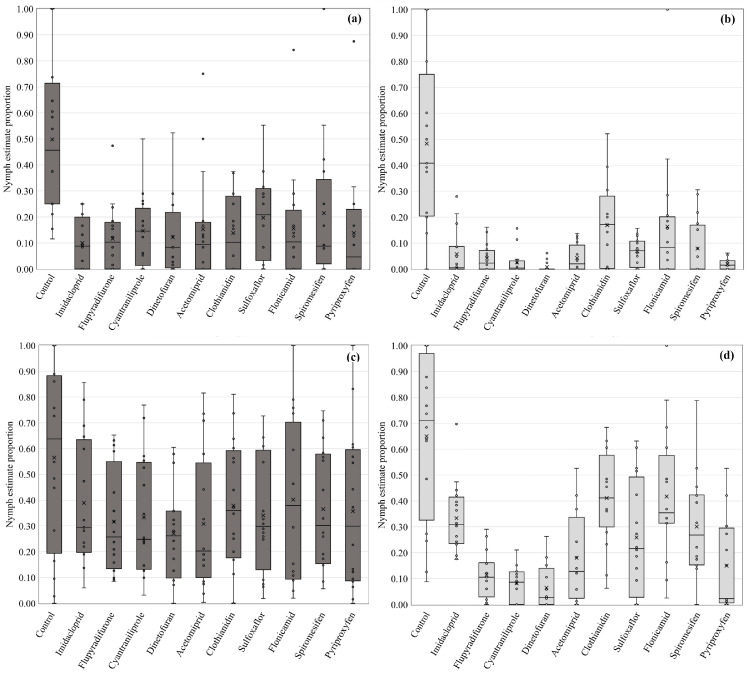
Proportional estimate of Bemisia tabaci nymph numbers from field and laboratory insecticide efficacy trials. (**a**) field estimate from the 2021 season; (**b**) laboratory bioassay estimate from the 2021 season; (**c**) field estimate from the 2022 season; (**d**) laboratory bioassay estimate from the 2022 season. The box plot includes the interquartile range (IQR, the box), the median (line in the box), data variability (whiskers), overall mean (X), and trial/bioassay means (dots).

**Figure 3 insects-14-00645-f003:**
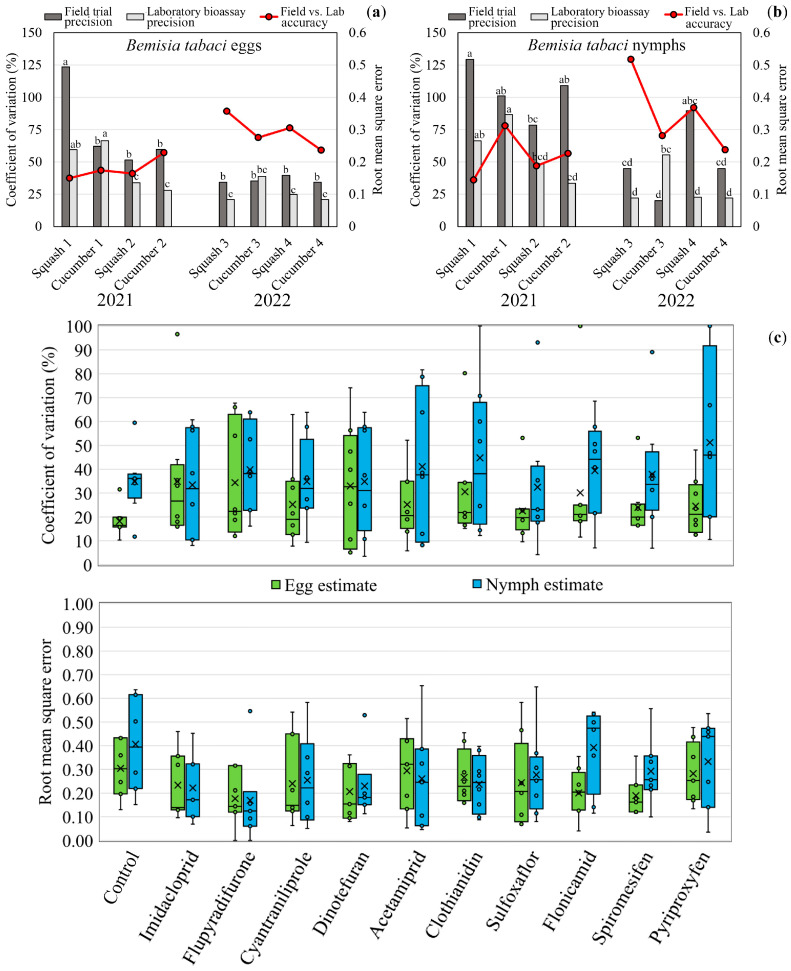
Precision and accuracy measures of Bemisia tabaci eggs and nymphs treated with insecticides in field and laboratory. (**a**) Coefficient of variation (columns) and root mean square error (lines) of egg numbers throughout the field and laboratory experiments; (**b**) Coefficient of variation (columns) and root mean square error (lines) of nymph numbers throughout the field and laboratory experiments (**c**) Bioassay precision (coefficient of variation) and accuracy (root mean square error) estimates distribution by insecticide treatment. Different letters in the columns with same color indicate significant differences, Tukey’s test (*p* < 0.05). The box plot includes the interquartile range (box), the median (line in the box), data variability (whiskers), overall mean (X), and trial/bioassay means (dots).

**Figure 4 insects-14-00645-f004:**
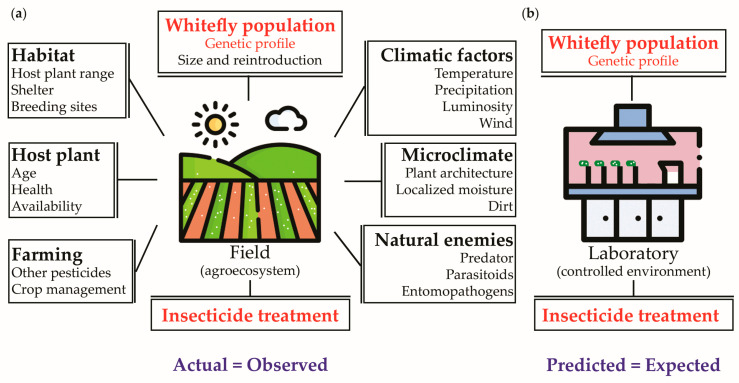
Schematic model representing factors contributing to statistical error in field and laboratory conditions, and the definitions of accuracy variables (bottom) for the experimental comparison of insecticide efficacy of *Bemisia tabaci* immatures in the two venues. Note that (**a**) represents the field contributing factors and their definitions; while (**b**) represents the greatly simplified conditions of a laboratory bioassay.

**Table 1 insects-14-00645-t001:** Description of insecticide treatments (™ = trademark) based on the maximum commercial labeled rate for controlling *Bemisia tabaci* in field and laboratory experiments. Tifton, GA, 2021–2022.

IRAC ^1^ Group	Common Name	Commercial™ Name	Per Hectare Rate ^2^
-	Water check	-	-
4A	Imidacloprid	Admire Pro 4.6F	160.8 mL
4A	Dinotefuran	Venom 70SG	280.2 g
4A	Acetamiprid	Assail 30SG	280.2 g
4A	Clothianidin	Belay 50WDG	292.3 mL
4C	Sulfoxaflor	Transform WG	157.6 g
4D	Flupyradifurone	Sivanto Prime 1.67SL	876.9 mL
7C	Pyriproxyfen	Knack 0.86EC	730.8 mL
9C	Flonicamid	Beleaf 50SG	299.8 g
23	Spiromesifen	Oberon 2SC	621.1 mL
28	Cyantraniliprole	Exirel 0.83SC	986.5 mL

^1^ Insecticide Resistance Action Committee; ^2^ Considering total volume of 935 L ha^−1^ (equiv. 100 gal. acre^−1^).

## Data Availability

The data presented in this study are available at the request of the corresponding authors.

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
