# Peer review of "Precision and Accuracy of Field Versus Laboratory Bioassay Insecticide Efficacy for the Control of Immature Bemisia tabaci"

_insects, 2023, doi:10.3390/insects14070645_

Round 1

Reviewer 1 Report

Document is well written. In line 235, change larval by nymphal or immatures. You lost the focus in the discussion. Concentrate in: differences in precision and accuracy in the different evaluated insecticides and their possible use as predictors of a reliable field test, as well as for possible resistance monitoring. You performed your study for such reasons.

Author Response

We sincerely appreciate the reviewer`s time and effort spent reviewing our manuscript and your insightful comments.

In line with your feedback, we have made the suggested comment in line 236 (updated after corrections), changing "larval" to "nymphal".

Regarding your point about the discussion, we have taken your advice on board. We improved our discussion to highlight the differences in precision and accuracy among the evaluated insecticides and their potential use as predictors of a reliable field test, as well as for possible resistance monitoring. We include a recently published paper from our group on laboratory-field adult efficacy determination. This paper was not published at the time of our initial submission, and we believe it significantly strengthens our discussion and conclusions.

Reviewer 2 Report

Ecotoxicological studies often result in reports on the limitation and sometime failures of  biological assay data to predict field response to similar treatments. To quantify the relationship between whitefly bioassay and field response data, the authors evaluated a controlled laboratory bioassay and a concurrent cucurbit field trial method to assess insecticide efficacy for controlling the sweetpotato whitefly. And the bioassays provide significantly precision for estimating proportional whitefly response. Overall, this manuscript provide very useful information for detertermine the insecticides response. 

Author Response

The authors are thankful for the encouraging feedback on our work.

Reviewer 3 Report

This manuscript evaluates the precision and accuracy of laboratory bioassays performed in assessing the efficacy of commonly used insecticides against Bemisia tabaci in comparison with efficacy results and monitoring the development of resistance to the same insecticides determined in the field.

The laboratory and field trials were adequately designed and conducted. The manuscript is well-written with carefully organized text. The results are sufficiently presented and analyzed, and the discussion is valuable for a better understanding the findings.

Therefore, the manuscript is recommended for publication in the Insects, after correcting two minor oversights in lines 379 and 414, where the scientific names of the insects should be written in italics.

Author Response

We appreciate the reviewer`s time and effort in correcting our manuscript. This positive feedback and constructive comments are highly appreciated.

Regarding the minor oversights in lines 393 and 430 (updated after corrections), we have corrected the scientific names in italics as per the standard conventions.

Round 2

Reviewer 1 Report

You will receive comments from the main editor.

Author Response

We appreciate the reviewer time and effort to improve our manuscript. We address the comments from the main editor and made all necessary revisions to improve our work.